# Tanshinone functions as a coenzyme that confers gain of function of NQO1 to suppress ferroptosis

Tian-Xiang Wang[1],[*] , Kun-Long Duan[1],[*] , Zi-Xuan Huang[1], Zi-An Xue[1], Jun-Yun Liang[1], Yongjun Dang[2], Ao Zhang[3], Yue Xiong[4], Chunyong Ding[3] , Kun-Liang Guan[5], Hai-Xin Yuan[1],[2]

**Ferroptosis is triggered by the breakdown of cellular iron-dependent redox homeostasis and the abnormal accumulation of lipid ROS. Cells have evolved defense mechanisms to prevent lipid ROS accumulation and ferroptosis. Using a library of more than 4,000 bioactive compounds, we show that tanshinone from *Salvia miltiorrhiza* (Danshen) has very potent inhibitory activity against ferroptosis. Mechanistically, we found that tanshinone functions as a coenzyme for NAD(P)H:quinone oxidoreductase 1 (NQO1), which detoxifies lipid peroxyl radicals and inhibits ferroptosis both in vitro and in vivo. Although NQO1 is recognized as an oxidative stress response gene, it does not appear to have a direct role in ferroptosis inhibition in the absence of tanshinone. Here, we demonstrate a gain of function of NQO1 induced by tanshinone, which is a novel mechanism for ferroptosis inhibition. Using mouse models of acute liver injury and ischemia/reperfusion heart injury, we observed that tanshinone displays protective effects in both the liver and the heart in a manner dependent on NQO1. Our results link the clinical use of tanshinone to its activity in ferroptosis inhibition.**

## Introduction

Ferroptosis is a nonapoptotic form of regulated cell death triggered by the breakdown of cellular iron-dependent redox homeostasis and the abnormal accumulation of lipid ROS (Yang & Stockwell, 2016). Glutathione-dependent lipid ROS elimination by GPX4 is considered to be the primary mechanism that defends against excessive lipid ROS accumulation (Friedmann Angeli et al, 2014; Yang et al, 2014). Recently, pathways parallel to GPX4, such as the FSP1/vitamin K, FSP1/DHODH-CoQ$_{10}$, and DHFR/BH4 systems, were found to trap lipid peroxyl radicals and suppress ferroptosis (Bersuker et al, 2019; Doll et al, 2019; Soula et al, 2020; Mao et al, 2021; Mishima et al, 2022). Among these new pathways, the enzyme-mediated generation of

lipophilic radical-trapping antioxidant (RTA), which halts the propagation of lipid peroxides, is the core step.

However, during pathological processes, the abundant generation of lipid ROS or damage to cellular defense pathways ultimately results in ferroptosis (Stockwell et al, 2020). Ferroptosis-targeted therapy may shed light on pathological processes such as ischemia/reperfusion injury.

Here, we report tanshinone family compounds act as potent ferroptosis inhibitors. Tanshinone is the lipid-soluble active ingredient of Danshen, which is one of the most widely used traditional Chinese medicines, mainly for protection against cardiovascular diseases (Chen & Chen, 2017; MEIm et al, 2019), but the molecular basis for its mechanism of action is largely unknown. Through a combination of thermal shift experiments and label-free mass spectrum analysis, we found that NQO1 is the direct molecular target of tanshinone. Taking advantage of a series of tanshinone derivatives and in vitro enzyme activity assays, we further demonstrated that tanshinone functions as a coenzyme for NQO1 by accepting electrons from FAD to generate reduced tanshinone at the expense of NADH, which in turn detoxifies lipid peroxyl radicals and inhibits ferroptosis. In this regard, tanshinones display extremely high potency because they act catalytically in quenching lipid ROS.

To further link the clinical use of tanshinone to its activity in ferroptosis inhibition, we generated NQO1 KO mice and found that tanshinone displays protective effects in both the liver and the heart in a manner dependent on NQO1. Our study reveals a molecular mechanism for the therapeutic benefit of tanshinone, such as cardiac protection from ischemia.

## Results

### Tanshinone inhibits ferroptotic cell death

To identify novel ferroptosis inhibitors, we screened a library of more than 4,000 compounds (Li et al, 2019), consisting of FDA-

[1]The Fifth People's Hospital of Shanghai, The Molecular and Cell Biology Research Lab of the Institutes of Biomedical Sciences and the School of Pharmacy, Fudan University, Shanghai, China    [2]Center for Novel Target and Therapeutic Intervention, Chongqing Medical University, Chongqing, China    [3]Pharm-X Center, College of Pharmaceutical Sciences, Shanghai Jiao Tong University, Shanghai, China    [4]Cullgen Inc., San Diego, CA, USA    [5]Department of Pharmacology and Moores Cancer Center, University of California San Diego, La Jolla, CA, USA

Correspondence: yuanhaixin@fudan.edu.cn; chunding@sjtu.edu.cn
*Tian-Xiang Wang and Kun-Long Duan contributed equally to this work

approved drugs and natural compounds (Fig 1A). Ferroptosis inhibition was assayed in HT-1080 fibrosarcoma cells that were treated with the ferroptosis inducer erastin. Among the positive candidates are several known ferroptosis inhibitors (Dixon et al, 2012), including iron chelators, lipoxygenase inhibitors, and RTA (Fig 1A), thus validating the screening strategy. Notably, a group of tanshinone family compounds showed potent ferroptosis inhibitory activity, including tanshinone I, tanshinone IIA, and cryptotanshinone (Fig 1A and B).

Tanshinone compounds are the major lipid-soluble components of the traditional Chinese herbal Danshen, which is very popular. Tanshinone IIA, cryptotanshinone, and tanshinone I inhibited erastin-induced ferroptosis with EC50 values of 7, 9, and 30 nM, respectively (Fig 1B). Similar activity was observed against sulfasalazine- or RSL3-induced ferroptosis (Fig 1C). In addition, cryptotanshinone also inhibited ferroptosis induced by glutamate, sulfasalazine, or RSL3 in HT-22 and C2C12 cells, suggesting a broad effect of tanshinone in terms of ferroptosis suppression (Fig 1D).

As excessive lipid ROS is widely recognized as the critical cause of ferroptosis, we investigated whether tanshinone could suppress lipid ROS accumulation. HT-1080 cells were treated with erastin for 10.5 h to induce lipid ROS accumulation followed by the addition of tanshinone IIA, cryptotanshinone, or the positive control Trolox, which is a water-soluble derivative of vitamin E with potent reducibility and is widely used to inhibit ferroptosis. All three compounds efficiently reduced lipid ROS levels (Fig 2A). This result suggests that tanshinone functions during the process of lipid ROS elimination but not accumulation. However, unlike Trolox, tanshinones did not directly react with the free radical DPPH (2,2-diphenyl-1-picrylhydrazyl) (Fig 2B) or ABTS (2,2′-azino-bis (3-ethylbenzthiazoline-6-sulfonic acid)) (Fig 2C), indicating that tanshinones do not serve as RTAs. Collectively, our observations imply that endogenous factors may endow tanshinones with lipid ROS elimination activity. Moreover, the effect was specific to lipid ROS elimination, as tanshinone IIA failed to inhibit $H_2O_2$-induced necrosis (Fig 2D).

The GSH-dependent elimination of lipid ROS by GPX4 is essential for ferroptosis suppression. To explore whether the effect of tanshinone requires GPX4, we generated GXP4 KO HT-1080 cells (Fig 2E), which underwent ferroptosis in the absence of the ferroptosis inhibitor ferrostatin-1 (Fer-1) (Yang et al, 2014). Interestingly, tanshinone IIA and cryptotanshinone inhibited ferroptosis in GPX4 KO cells with EC50 values similar to those in WT cells (Fig 2E), indicating that tanshinone acts independently of GPX4.

### Tanshinones function as a coenzyme for NQO1 to reduce lipid ROS

To understand the mechanism of tanshinones in ferroptosis inhibition, we examined more tanshinone derivatives (Fig 3A). As shown in Fig 3B, most of the compounds showed efficient inhibitory activity against ferroptosis at concentrations as low as 50 nM, and more specifically, Compound 15-047 had the highest efficiency (EC50 ≈ 2 nM). In contrast, Compounds 15-495 and 15-422 exhibited much worse activity (Fig 3C).

To identify the molecular target(s) that empower tanshinones with lipid ROS elimination activity, we performed a thermal shift experiment using Compound 15-047 followed by label-free proteomic analysis based on the hypothesis that compound binding may stabilize or destabilize target proteins. Interestingly, 15-047 treatment only altered the stability of a few proteins (Figs 4A and S1A). The NAD(P)H:quinone dehydrogenase (NQO1) was most notable because of its role in redox reactions. We confirmed that 15-047 reduced NQO1 thermal stability and that its melting temperature (Tm) was decreased by ~4°C (Fig 4B and C).

NQO1 is a quinone reductase. Interestingly, tanshinones have orthoquinone structures, which can be reduced by NQO1. We propose that tanshinone reduction by NQO1 may serve as a lipophilic RTA to eliminate lipid ROS. A docking model of NQO1 in complex with 15-047 and FAD was constructed to illustrate the possible structure–activity relationship (Fig 4D). The binding mode implies that 15-047 may serve as an NQO1 substrate that accepts electrons from FAD to generate the reduced form of 15-047. The docking model also explains why 15-495 or 15-422 shows little ferroptosis inhibitory ability, as there is not enough room for the bulk groups near the furan ring in these two compounds. To confirm tanshinones as substrates of NQO1, an in vitro NQO1 enzyme activity assay was performed. In the presence of 15-047 (Fig 4E and F) or tanshinone IIA (Fig S1B), NADH was rapidly consumed in an NQO1 dose-dependent manner. In contrast, FSP1, which uses coenzyme Q as a substrate, showed little catalytic activity toward tanshinone (Fig 4E). In addition, in line with their poor activity in suppressing ferroptosis, Compounds 15-495 and 15-422 failed to promote NADH consumption, indicating that they are not NQO1 substrates (Fig 4F).

To further test our hypothesis that reduced tanshinones are responsible for lipid ROS elimination, we carried out the co-autoxidation of egg phosphatidylcholine and C11-BODIPY581/591 using the radical generator 2,2′-azobis (2-methylpropionamidine) dihydrochloride (AAPH) (Yoshida et al, 2003). Neither NQO1 alone nor the combination of NQO1 and NADH effectively suppressed the accumulation of lipid ROS, whereas the addition of tanshinone IIA significantly retarded autoxidation (Fig 4G). Similarly, 15-047 also slowed autoxidation in a dose-dependent manner (Fig 4H). These results demonstrate that through NQO1, tanshinones relay reducing equivalents from NAD(P)H to inhibit the propagation of lipid peroxidation.

### NQO1 is required for the ferroptosis protective activity of tanshinones

To test the function of NQO1 in tanshinone-induced ferroptosis inhibition, we generated NQO1 KO HT-1080 cell lines (Fig 5A), which had no effect on the expression of ferroptosis-related genes (Fig S2A and B). NQO1 KO abolished the protective effect of tanshinone IIA against ferroptosis (Fig 5A). The reconstitution of NQO1 in KO cells fully restored the function of tanshinone IIA in ferroptosis inhibition (Fig 5B). We also observed that the ferroptosis inhibitory activity of compounds 15-045, 14-324, and 15-047 was abolished in NQO1 KO cells (Fig 5C), indicating a general role of NQO1 in mediating tanshinone function. Nqo1 KO in C2C12 cells similarly abolished the ferroptosis inhibitory effect of 15-047 (Fig 5D).

Notably, when a very high concentration was tested, 15-047 exhibited a ferroptosis inhibitory effect in NQO1 KO cells (Fig 5E and F), suggesting that other enzymes may catalyze the reduction of tanshinone. However, NQO2 KO did not alter the protective effect of tanshinone IIA (Fig S3A and B). Furthermore, the inhibition of FSP1

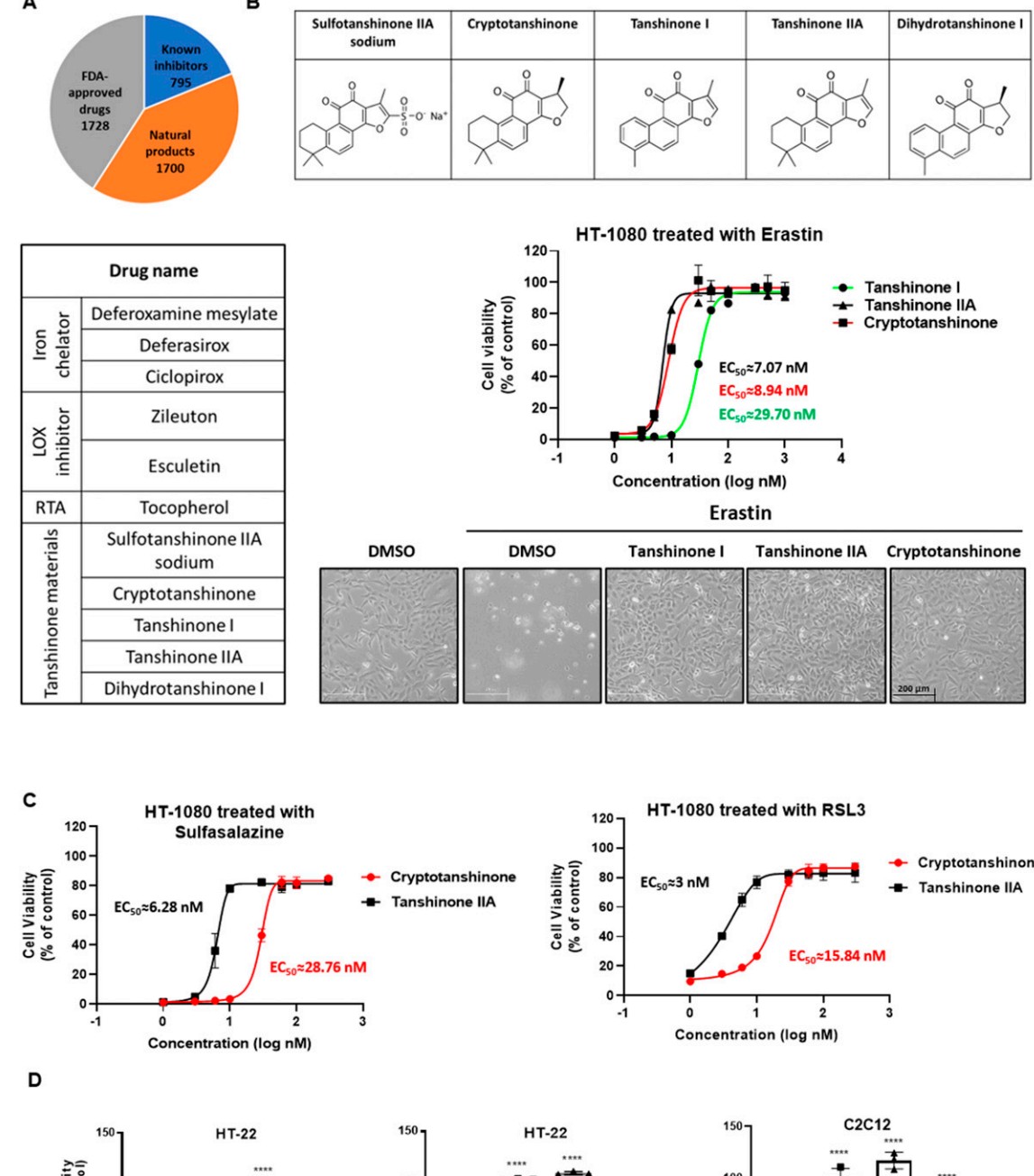

**Figure 1. Tanshinone compounds potently inhibit ferroptosis.**
**(A)** High-throughput screening for ferroptosis inhibitors. HT-1080 cells were treated with erastin (10 $\mu M$) and one chemical (20 $\mu M$) for 24 h, and then, cell viability was assayed. Positive chemicals including iron chelators, lipoxygenase inhibitors, and radical-trapping antioxidant molecules are known ferroptosis inhibitors. Tanshinone family compounds are molecules of interest. **(B)** Tanshinones inhibit erastin-induced ferroptosis in HT-1080 cells. Structures of five tanshinone compounds are shown in the top panel; ferroptosis protection curves are shown in the middle panel, and data are presented as the mean ± S.D., n = 3 independent repeats; cell morphology is shown in the bottom panel (50 nM tanshinones are used respectively). **(C)** Cryptotanshinone and tanshinone IIA inhibit RSL3- and sulfasalazine-induced ferroptosis. HT-1080 cells were treated with RSL3 (1 $\mu M$) or sulfasalazine (400 $\mu M$) to induce ferroptosis. Different concentrations of cryptotanshinone or tanshinone IIA were added simultaneously. Cell viability was assayed after 24-h treatment. Data are presented as the mean ± S.D., n = 3 independent repeats. EC50 values were calculated by a

did not influence the inhibitory effect of tanshinone (Fig 5G). Interestingly, we observed that NQO1 deficiency did not affect cell sensitivity to erastin or RSL3 (Fig 5H), whereas the overexpression of FSP1, but not NQO1, reduced cell sensitivity to ferroptosis (Fig 5I). These results indicate that the function of NQO1 in ferroptosis is strictly dependent on tanshinone compounds.

### Tanshinones protect against ferroptosis-related pathology in a manner dependent on NQO1

We posited that the ferroptosis inhibitory activity of tanshinones represents the biochemical basis for the therapeutic benefit of these compounds. To test this hypothesis, *Nqo1* KO mice were generated, and Nqo1 deficiency in multiple tissues was verified by Western blot (Fig 6A). Acute liver injury (ALI) induced by concanavalin A (ConA) has been reported to be a model of ferroptotic injury (Yan et al, 2021). We thus used this model to test our hypothesis. DMSO or 15-047 was intraperitoneally injected into WT and *Nqo1* KO mice, followed by the injection of sublethal doses of ConA to induce ALI. As expected, serum ALT and AST levels were significantly elevated 24 h after ConA injection, and these liver injury markers were significantly reduced by 15-047 treatment (Fig 6B). These results are consistent with the role of ferroptosis in ConA-induced ALI, which could be protected by tanshinones.

In *Nqo1* KO mice, 15-047 was less effective in suppressing ConA-induced ALT and AST (Fig 6B). Notably, the loss of Nqo1 did not fully abolish the activity of 15-047 against ALI, indicating that in addition to Nqo1, other components may contribute to 15-047's activity in the ALI model. Consistently, histological analysis showed that 15-047 pretreatment decreased the ConA-induced necrotic area in the liver, whereas Nqo1 deficiency suppressed the protective effect of 15-047 (Fig 6C). As a control, Fer-1 suppressed ConA-induced ALI regardless of the NQO1 status (Fig 6C). These data indicate that tanshinone protects against liver ferroptosis injury, which is at least in part dependent on NQO1.

Ferroptosis is believed to play an important role in ischemia/reperfusion (I/R) organ injury (Friedmann Angeli et al, 2014; Gao et al, 2015; Fang et al, 2019). Sulfotanshinone sodium injection has been widely used as an adjuvant therapy for ischemia heart injury and cardiovascular protection (Zhou et al, 2019). We thus adopted the I/R heart injury model to test the tanshinone–NQO1 hypothesis. Mice were subjected to surgery for 45 min/24 h of I/R treatment. The operation group showed the loss of cardiac function compared with the sham group as demonstrated by lower left ventricular ejection fraction and left ventricular fraction shortening (Fig S4). The pretreatment of WT mice with 15-047 reduced the myocardial infarct size compared with the DMSO group, although these groups had comparable areas at risk (Fig 6D). Importantly, the protective effect of 15-047 was largely abolished in *Nqo1* KO mice. Consistent with these observations, 15-047 pretreatment also alleviated I/R-induced up-regulation of myocardial enzymes in the serum, including lactate dehydrogenase (LDH), creatine kinase, and cardiac troponin I (cTnI).

Again, Nqo1 deficiency significantly abolished the protective effect of 15-047 (Fig 6E and F). Moreover, Nqo1 KO also suppressed the protection of cardiac function by 15-047 (Fig S4). Taken together, these mouse data support that NQO1-catalyzed tanshinone reduction represents a key mechanism for the protective function of tanshinones against ferroptosis-induced pathology.

## Discussion

In this study, we demonstrated that tanshinones are potent ferroptosis inhibitors both in vitro and in vivo. The low nanomolar EC50 of tanshinone in suppressing ferroptosis makes it among the most potent ferroptosis inhibitors known to date. Mechanistically, the orthoquinone moiety in tanshinone could be reduced by NQO1 to become hydroxyl orthoquinone (Hao et al, 2007), which can serve as an electron donor to reduce lipid ROS, and with the concomitant oxidation of hydroxyl orthoquinone to the original orthoquinone moiety in tanshinone, this completes the tanshinone catalytic cycle at the expense of NAD(P)H. This mechanism also explains the extremely high efficacy of tanshinone in protecting cells from ferroptosis, and the requirement for NQO1 in this process.

NQO1 is a classic NAD(P)H dehydrogenase that catalyzes the two-electron reduction of quinones and a wide range of other organic compounds (Pey et al, 2019). As a target of Nrf2, NQO1 is often up-regulated in response to oxidative stress and contributes to oxidative protection (Dinkova-Kostova & Talalay, 2010). NQO1 is reported to be capable of catalyzing the reduction of CoQ to CoQH$_2$ (De Cabo et al, 2004; Ross & Siegel, 2017), similar to FSP1. However, NQO1 does not appear to have a direct role in ferroptosis inhibition in the absence of tanshinones because neither NQO1 KO nor its overexpression influences ferroptosis sensitivity. Thus, under physiological conditions, NQO1 may not be involved in ferroptosis regulation. We propose that tanshinones confer a new gain of function of NQO1, as the combination of tanshinone and NQO1 provides potent protection against ferroptosis.

Tanshinones have been broadly studied for their antitumor activity by inducing cell death (Chen et al, 2014; Li et al, 2020). Interestingly, NQO1 has been implicated in the anti-apoptotic activity of tanshinones (Zhong et al, 2021). At first glance, these observations appear at odds with our results. However, these studies often used micromolar concentrations of tanshinones, much higher than their nanomolar concentrations, for ferroptosis inhibition. We also observed that the overexpression of NQO1, but not NQO2, augmented the cytotoxicity of tanshinones at concentrations of 5–10 $\mu$M (Fig S5). The cell death induced by high concentrations of tanshinone was not ferroptosis, as it could not be suppressed by DFO (Fig S5). The hydroquinone produced by NQO1 could undergo autoxidation to yield H$_2$O$_2$, which can impose oxidative stress on cells (Ross & Siegel, 2017). In addition, tanshinone reduction by NQO1 consumes NAD(P)H. A high concentration of tanshinone may lead to the exhaustion of cellular NAD(P)H and the accumulation of H$_2$O$_2$, thus sensitizing cells to oxidative stress (Shimada et al, 2016).

---

nonlinear regression model (log[agonist] versus response–variable slope [four parameters]). **(D)** Cryptotanshinone inhibits ferroptosis induced by glutamate (10 mM), sulfasalazine (400 $\mu$M), or RSL3 (1 $\mu$M) in HT-22 and C2C12 cells. Cell viability was measured after 24-h treatment (n = 3; ****$P$ < 0.0001 versus DMSO group, Dunnett's multiple comparisons test).

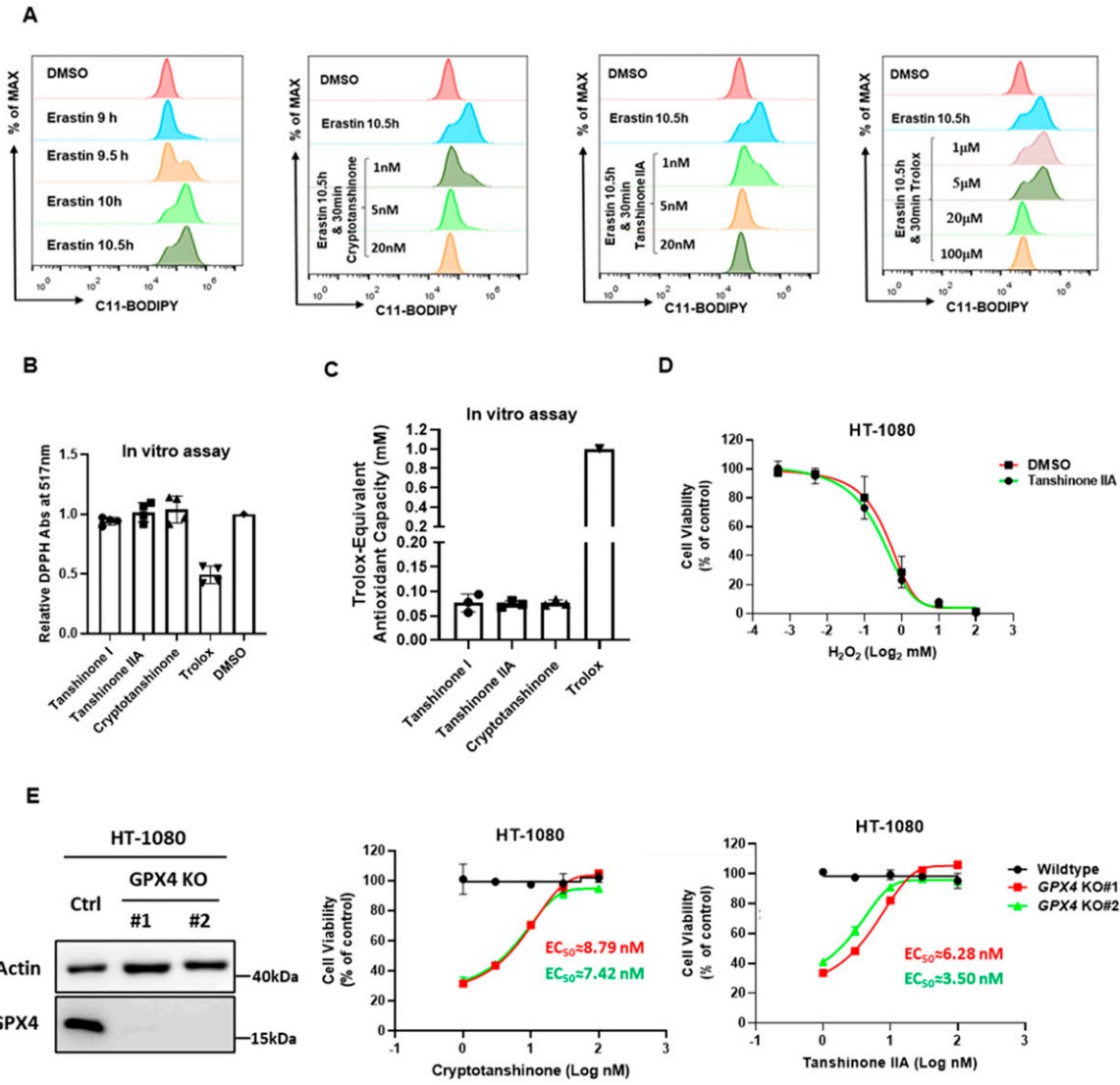

**Figure 2. Tanshinone compounds inhibit ferroptosis through eliminating accumulated lipid ROS in a GPX4-independent manner.**
**(A)** Tanshinones eliminate lipid ROS. HT-1080 cells were treated with erastin for 10.5 h followed by the addition of cryptotanshinone, tanshinone IIA, or Trolox for an extra 30 min. Cellular lipid ROS levels were assessed by C11-BODIPY581/591 staining coupled with flow cytometry analysis. **(B)** Tanshinones do not directly react with 2, 2-diphenyl-1-picrylhydrazyl. The stable radical 2, 2-diphenyl-1-picrylhydrazyl was dissolved in methanol and then incubated with a test compound dissolved in DMSO. Trolox was included as a positive control. The absorbance at 517 nm was recorded and normalized to background (methanol only). **(C)** Tanshinones show little antioxidant capacity in vitro. The antioxidant capacity was calculated by the 2,2'-azino-bis 3-ethylbenzthiazoline-6-sulfonic acid method. The absorbance at 734 nm was recorded, and Trolox equivalent antioxidant capacity was calculated. **(D)** Tanshinone IIA does not inhibit $H_2O_2$-induced cell death. HT-1080 cells incubated with or without tanshinone IIA (50 nM) were treated with different concentrations of $H_2O_2$ for 24 h, and cell viability was assayed. **(E)** Cryptotanshinone and tanshinone IIA protect cells from GPX4 deficiency–induced cell death. Validation of GPX4 KO in HT-1080 cells by Western blot. HT-1080 cells with/without GPX4 KO were treated with cryptotanshinone or tanshinone IIA for 24 h, and cell viability was assayed. Data are presented as the mean ± S.D., n = 3 independent repeats. EC50 values were calculated by a nonlinear regression model (log[agonist] versus response–variable slope [four parameters]).

Collectively, our findings reveal that the tanshinone–NQO1 reaction system can either protect cells from ferroptosis or harm cells by oxidative stress depending on the concentration of tanshinones. This knowledge will be valuable for selecting medical doses of tanshinone in clinical applications. Future research must carefully consider the compound concentration factor.

Numerous studies suggest that the pharmacological modulation of ferroptosis may yield significant clinical benefits for certain diseases, such as ischemic organ injuries (Stockwell et al, 2020). Using

mouse models, we demonstrated that 15-047 alleviated the pathological process of ConA-induced ALI and ischemia/reperfusion-induced heart injury. These pharmacological activities of tanshinone are at least partly dependent on NQO1. We posited that the tanshinone–NQO1 connection provides a molecular mechanism for the therapeutic effects of tanshinones, such as cardiac protection.

Danshen, as traditional Chinese medicine, has been used for thousands of years in East and South Asia. Currently, hundreds of

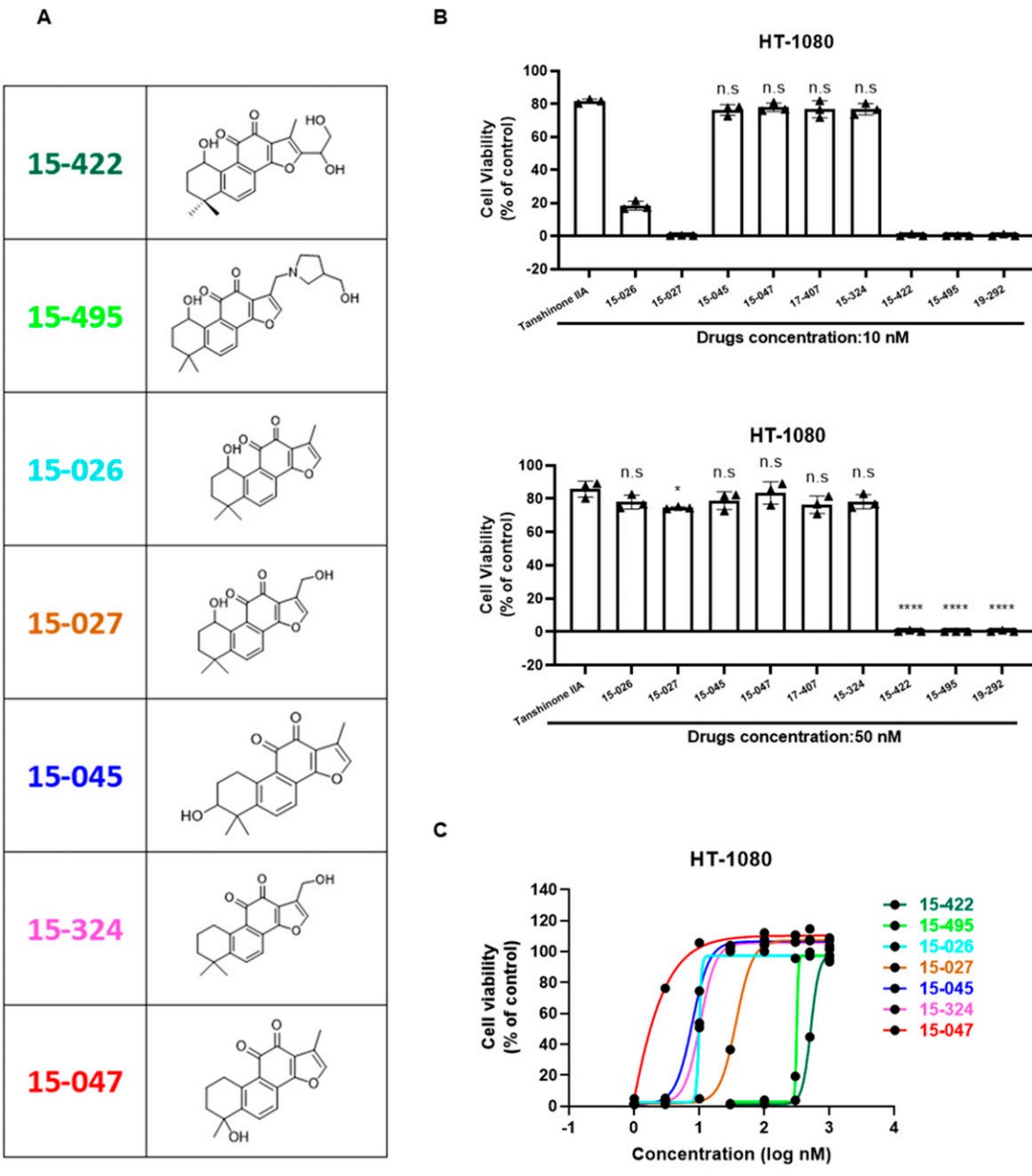

**Figure 3. Identification of 15-047 as the most effective ferroptosis inhibitor derived from tanshinone IIA.**
**(A)** Structures of tanshinone derivatives. **(B)** Tanshinone derivatives showed different inhibitory effects on ferroptosis. HT-1080 cells incubated with erastin were treated with individual tanshinones (10 or 50 nM). Cell viability was assayed after 24-h treatment (n = 3; ****P < 0.0001 versus tanshinone IIA group, Dunnett's multiple comparisons test). **(C)** Activity of tanshinone derivatives against ferroptosis. HT-1080 cells were treated with erastin and different concentrations of the tanshinone derivatives as shown in (A). Cell viability was assayed after 24-h treatment. Data are presented as the mean ± S.D., n = 3 independent repeats.

millions of doses of Danshen are used annually to treat angina and coronary heart diseases by more than 10 million patients around the world (Mader, 2010). One of the major Danshen medicines, Danshen Dripping Pill, was sold for three billion yuan in 2020 according to a financial report (Tasley, 2020). Inspired by the extensive usage of Danshen, we propose that the tanshinone–NQO1 mechanism should be considered for the therapeutic studies of tanshinones in many disease models (Stockwell et al, 2017; Wang et al, 2020).

# Materials and Methods

### Chemicals and reagents

Cryptotanshinone, tanshinone I, tanshinone IIA, erastin, RSL3, sulfasalazine, Trolox, and cisplatin were purchased commercially from MedChemExpress (MCE). NADH and $H_2O_2$ were purchased from Beyotime Biotechnology. L-$\alpha$-phosphatidylcholine (egg, chicken)

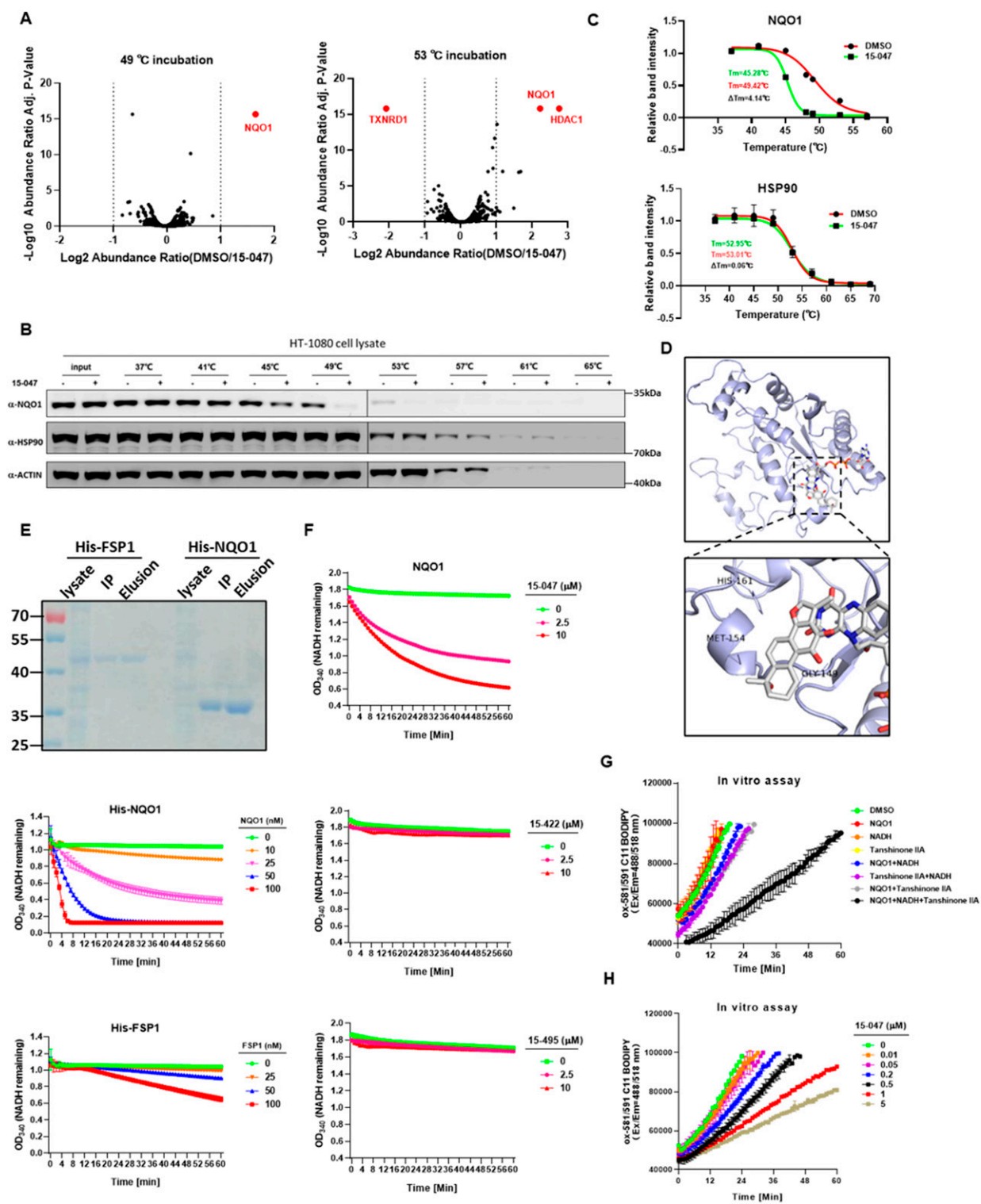

**Figure 4. Tanshinone compounds bind with NQO1 and eliminate lipid ROS at the expense of NADH.**
**(A)** Effect of 15-047 on protein stability under thermal stress. HT-1080 cells were treated with DMSO or 15-047 for 1 h, and then, cell lysates were incubated at 49°C or 53°C for 3 min. After clearance by centrifuge, the supernatants were subjected to label-free proteomic analysis. NQO1 protein was among the most significantly decreased proteins with 15-047 treatment. **(B)** 15-047 reduced NQO1 stability at thermal stress conditions. The thermal stability of NQO1, actin, and HSP90 with DMSO or 15-047 treatment was assayed by Western blot. **(C)** Melting temperatures (Tm) of NQO1 and HSP90 with DMSO or 15-047 treatment were calculated. The melting temperature shift (Δ Tm) was seen as a readout for compound binding affinity. **(D)** Docking model of NQO1 in complex with 15-047 and FAD. The model was generated by Molegro Virtual Docker to illustrate the structure–activity relationship between NQO1 and 15-047. The image was visualized by PyMOL. **(E)** NQO1, but not FSP1, catalyzes NADH consumption in the presence of 15-047. NADH (OD$_{340}$) consumption assay in PBS buffer using recombinant purified FSP1 or NQO1 in combination with 15-047. **(F)** 15-047, but not 15-422

was purchased from Merck. AAPH was purchased from Sigma-Aldrich. C11-BODIPY581/591 was purchased from Invitrogen.

## Cultured cell lines

C2C12, HT-22, and HT-1080 cells (ATCC) were cultured in DMEM containing 4.5 g/l glucose and L-glutamine. HT-1080 GPX4 KO cells were grown in DMEM supplemented with 100 nM of Fer-1. All media were supplemented with 10% fetal bovine serum (Gibco), and all cell lines were grown at 37°C with 5% $CO_2$.

## High-throughput screening of ferroptosis inhibitors

A total of 500,000 HT-1080 cells were seeded into 96-well plates (Corning). The next day, a fresh medium with erastin or DMSO added was used to replace the old medium in a 96-well plate. Compounds were also stored in 96-well plates. A total of 50 compound plates were used in the screening. An automatic sample injector was applied to add compounds into cells, and every well was treated with one different compound. After 24-h treatment, viabilities of cells in 96-well plates were assayed. The iron chelator DFO was added to some wells serving as a positive control. The corresponding compounds that showed comparable viabilities to DFO treatment were decoded for further analysis.

## Animal models

### All mouse studies were carried out in accordance with the National Institutes of Health

Guidelines were approved by the Laboratory Animal Ethical Committee of Fudan University. The *Nqo1* KO mice were generated through nonhomologous recombination using the CRISPR/Cas9 technology by Shanghai Model Organisms Center, Inc., and all mice were maintained on a pathogen-free condition on a 12-h light/dark cycle from 6 AM to 6 PM and provided unrestricted access to food and water.

## I/R heart injury mouse model

Adult WT and *Nqo1* KO male C57BL/6J mice at ages of 8–10 wk were anesthetized with isoflurane, and the left anterior descending (LAD) coronary artery was reversibly ligated using a sterile 7-0 silk suture with a slipknot. Proper ligation was confirmed by the visual observation of the left ventricle wall turning pale. After 45 min of regional ischemia, the heart was allowed to reperfusion, leading to the loss of the discoloration of the myocardium distal to the ligation. 15-047 (0.2 mg/kg) was administered by intraperitoneal injection 2 h before surgery. Sham operation mice underwent the same procedure without the ligation of the LAD coronary artery.

## Measurement of area at risk and infarct area of I/R heart injury mice

After the 24-h reperfusion period, the mice were anesthetized with isoflurane and the LAD was tightly ligated. Evans blue dye (5% in saline) was injected through an external iliac vein. After being excised and rinsed in PBS, the heart was frozen over dry ice for 20 min and cut transversely into slices (5 slices per heart). The slices were then incubated with 2,3,5-triphenyltetrazolium chloride solution (1.5%) to visualize the unstained infarct region. The infarct area, area at risk, and nonischemic left ventricle area were measured and analyzed by ImageJ.

## Echocardiography

Mice were subjected to M-mode echocardiography using a Vevo 770 small-animal echocardiographic analysis system (Visual Sonics Inc.). Briefly, mice were anesthetized using 1–1.5% isoflurane administered by inhalation and the upper sterna and subxiphoid areas were shaved. Using a 30-MHz transducer, the cardiac cycle parameters were recorded by performing M-mode echocardiography of the left ventricle in the parasternal long-axis view. The stable beat rates of all animals indicated the level of anesthesia state during the operation (300–350 bpm). The function of the left ventricle was assessed by the calculation of left ventricular ejection fraction and left ventricular fraction shortening. All measurements were manually obtained by the same observer.

## Measurement of CK, LDH, and cTnI release in serum

Serum from mice was collected and frozen at –80°C. Serum CK and LDH concentrations were analyzed using the Catalyst DX analyzer (IDEXX) following the manufacturer's instructions. The cTnI content was determined using the Mouse cTnI ELISA kit (H149-2; Jiancheng Biotech) by a microplate reader.

## ConA-induced acute liver injury mouse models

Adult WT and *Nqo1* KO male C57BL/6J mice at ages of 8–10 wk were injected with ConA (15 mg/kg) via the tail vein to induce ALI. Mice were euthanized at 24 h post-injection. The serum samples were collected to measure serum ALT and AST levels. Liver samples were fixed in 4% paraformaldehyde and embedded into paraffin to slice. The slides were subjected to hematoxylin and eosin staining. Images were captured using an OLYMPUS digital camera (DP71). Damaged areas were measured and analyzed by ImageJ.

---

or 15-495, serves as a substrate for NQO1. NADH ($OD_{340}$) consumption assay in PBS buffer using recombinant purified NQO1 in combination with different concentrations of tanshinone derivatives 15-047, 15-422, or 15-495. **(G)** NADH-dependent 15-047 reduction by NQO1 suppresses the autoxidation of liposomes. Autoxidation of 581/591 C11-BODIPY (1 μM) and liposomes of egg phosphatidylcholine lipids (1 mM) suspended in PBS (pH 7.4) was initiated by adding 1 mM AAPH. Recombinant purified NQO1, NADH, and tanshinone IIA were added to the reaction individually or in combination, and fluorescence signal of C11-BODIPY was monitored. **(H)** 15-047 reduction by NQO1 suppresses liposome autoxidation in a dose-dependent manner. Different concentrations of 15-047 in combination with purified NQO1 and NADH were added into the autoxidation reaction mix mentioned in (G), and fluorescence signal of C11-BODIPY was monitored.

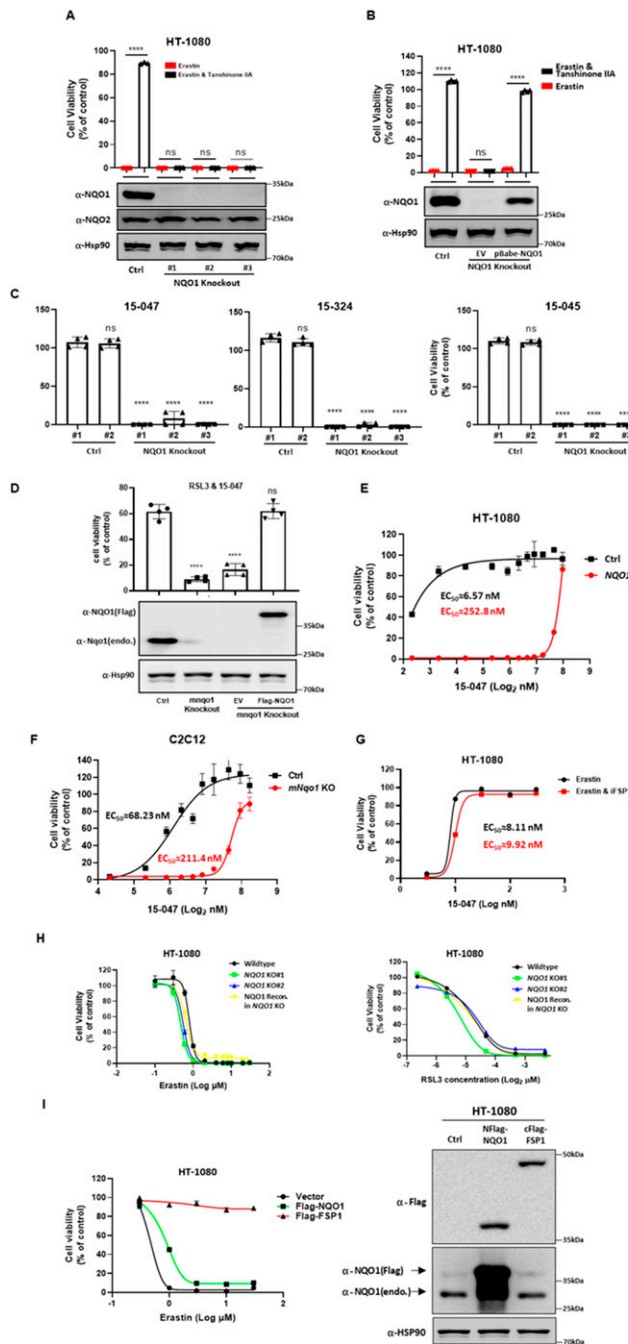

**Figure 5. NQO1 is required for the protective effect of tanshinone compounds against ferroptosis.**
**(A)** NQO1 KO abolishes the protective effect of tanshinone IIA against ferroptosis. WT and NQO1 KO HT-1080 cells were treated with erastin (10 $\mu$M) alone or together with tanshinone IIA (50 nM) for 24 h, and then, cell viability was assayed (n = 3; ****$P$ < 0.0001, Sidak's multiple comparisons test). **(B)** Reconstitution of NQO1 restores the inhibitory effect of tanshinone IIA on ferroptosis in NQO1 KO cells. HT-1080 cells with indicated NQO1 genotypes were treated with erastin (10 $\mu$M) alone or together with tanshinone IIA (50 nM) for 24 h, and then, cell viability was assayed (n = 4; ****$P$ < 0.0001, Sidak's multiple comparisons test). **(C)** NQO1 KO abolishes the protective effect of tanshinone compounds against ferroptosis. WT and NQO1 KO HT-1080 clone cells were treated with erastin and different tanshinone compounds (20 nM) for 24 h, and then, cell viability was assayed (n = 4; ****$P$ < 0.0001 versus Ctrl#1 group, Dunnett's multiple comparisons test). **(D)** NQO1 deficiency diminishes the protective effect of 15-047 against ferroptosis

## Cell viability assay

Cell viability was checked by measuring cellular ATP level using CellTiter-Lumi Plus Luminescent Cell Viability Assay Kit (Beyotime Biotechnology) according to the manufacturer's instructions. The luminescence intensity was read by a microplate reader (BioTek Synergy H1). All data were analyzed using GraphPad Prism (GraphPad Software, Inc.). The curves were fitted using a nonlinear regression model.

## The construction of CRISPR/Cas9 KO cells

Virus was produced by cotransfection of HEK293T cells with psPAX, pMD2.G, and pLenti expression plasmids at a 3:1:4 w/w/w ratio, using the Lipofectamine 2000 (Invitrogen) transfection reagent. The medium containing secreted virus was harvested after 36 h and sterile-filtered. CRISPR guide RNA (sgRNA) sequences targeting GPX4, NQO1, and NQO2 were designed using the online-available CRISPR design tool developed by the Zhang laboratory (http://crispr.mit.edu/). The sgRNA sequences were designed as follows:

hNQO1-sgRNA: *caccg* GTCCTTCAACTATGCCATGA;
mnqo1-sgRNA: *caccg* TGAATGGGCCAGTACAATCA;
NQO2-sgRNA: *caccg* ATATCTTTGTCTGTGGCCCT; and
GPX4-sgRNA: *caccg* CTTGGCGGAAAACTCGTGCA.

Nucleotides in italics show the overhangs introduced into oligos that are necessary for cloning into the BsmBI site of pLentiCRISPR v2.

Eighty cells were seeded into 96-well plates to get single-clone cells. After about 2 wk, grown-up single-clone cells were transferred to 24-well plates and gene expression in each clone was examined by Western blot.

## Exogenous gene expression

Cells stably expressing empty vector, Flag–NQO1, Flag–NQO2, or Flag–FSP1 were generated by lentiviral infection. In brief, HEK293T cells were

in C2C12 cells. C2C12 cells with Nqo1 KO or reconstitution were treated with RSL3 (1 $\mu$M) and 15-047 (100 nM) for 24 h, and cell viability was assayed (n = 4; ****$P$ < 0.0001 versus Ctrl group, Dunnett's multiple comparisons test). **(E)** Protective effect of 15-047 is dependent on NQO1 in HT-1080 cells. WT and NQO1 KO cells were treated with erastin alone or together with different concentrations of 15-047. Cell viability was assayed after 24-h treatment. **(F)** Protective effect of 15-047 is dependent on NQO1 in C2C12 cells. WT and *Nqo1* KO C2C12 cells were treated with erastin alone or together with different concentrations of 15-047. Cell viability was assayed after 24-h treatment. **(G)** Inhibition of FSP1 does not affect the protective effect of 15-047 against ferroptosis. HT-1080 cells incubated with erastin (10 $\mu$M) alone or together with iFSP1 (5 $\mu$M) were treated with different concentrations of 15-047 for 24 h, and then, cell viability was assayed. Data are presented as the mean ± S.D., n = 3 independent repeats. EC50 values were calculated by a nonlinear regression model (log[agonist] versus response–variable slope [four parameters]). **(H)** NQO1 deficiency does not affect cell sensitivity to ferroptosis. HT-1080 cells with NQO1 KO or reconstitution were treated with different concentrations of erastin or RSL3 for 24 h, and then, cell viability was assayed. **(I)** NQO1 overexpression does not affect cell sensitivity to ferroptosis. HT-1080 cells with the overexpression of Flag-tagged NQO1 or FSP1 were treated with different concentrations of erastin for 24 h, and then, cell viability was assayed. The expression of Flag-tagged proteins and endogenous NQO1 was detected by Western blot.

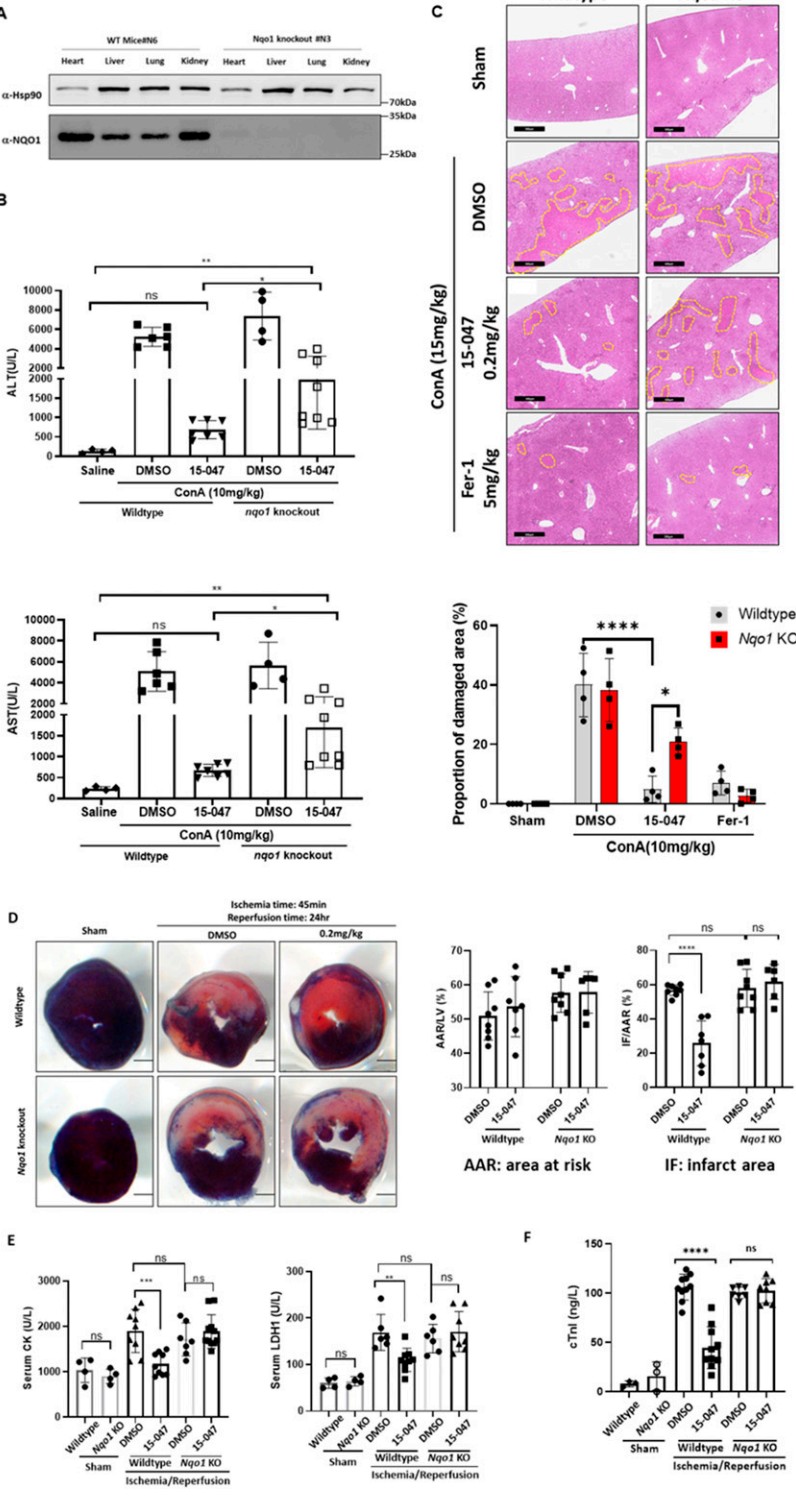

**Figure 6. Nqo1 mediates the protective effect of tanshinones against ferroptosis-related injury.**
**(A)** Whole-body *Nqo1* KO C57BL/6J mice were verified by Western blot analysis of different tissues. **(B)** Nqo1 deficiency dampens the protective effect of 15-047 on ConA-induced mouse liver injury. WT and *Nqo1* KO mice were treated with saline, ConA (10 mg/kg), or ConA with 15-047 (0.2 mg/kg) (n ≥ 4 mice for each group) for 24 h. Mouse serum was collected, and AST and ALT levels were measured. **(C)** Hematoxylin and eosin staining of liver sections from WT and *Nqo1* KO mice. Yellow dashed lines show damaged areas, which were quantified to calculate the ratio of the damaged area to total liver area (n = 4 mice for each group). **(D)** Nqo1 deficiency dampens the protective effect of 15-047 on mouse heart I/R injury. WT or *Nqo1* KO mice were injected with DMSO or 15-047 and then subjected to 45 min/24 h ischemia/ reperfusion injury. Mouse hearts were subjected to Evans blue dye staining and harvested for frozen section. The slices were then incubated with 2,3,5-triphenyltetrazolium chloride solution to visualize the unstained infarct region. Representative images and quantitative data of area at risk and infarct area in heart sections are shown (n ≥ 6 mice for each group). **(D, E)** Serum LDH1 and CK levels of WT or *Nqo1* KO mice as treated in (D) (sham group, n = 4; ischemia/reperfusion group, n ≥ 6). **(D, F)** Serum cardiac troponin I levels of WT or *Nqo1* KO mice as treated in (D) (sham group, n = 3; ischemia/reperfusion group, n ≥ 7).

transfected with the packaging plasmids pMD2.g and psPAX2, plus the pLVX vector, or with the packing plasmids VSVg and GAG, plus the pBABE vector ligated with the DNA fragments listed. 48 h after transfection, the viral supernatant was collected, purified through centrifugation and a 0.45-µm filter, and used to infect the indicated cells with Polybrene (5 µg/ml). 48 h after infection, positive cells were enriched with treatment of cells with puromycin (1–2 µg/ml) or hygromycin B (100–200 µg/ml).

### RNA isolation, reverse transcription–PCR, and quantitative real-time PCR

Cellular RNA was extracted with TransZol Up reagent (ET111-01; TransGen Biotech) and reverse-transcribed using EZscript Reverse Transcription Mix (EZB-RT2GQ; EZBioscience) following the manufacturer's instructions. Quantitative real-time PCR (qRT-PCR) was performed in QuantStudio 6 Flex (Applied Biosystems) using ChamQ Universal SYBR qPCR Master Mix (Q711-02; Vazyme). The relative abundance of mRNA was calculated by ΔCt values with actin mRNA as an internal control. *P*-values were obtained using a two-tailed *t* test, unless otherwise indicated in the figure legend. Primer sequences were listed as follows:

Homo-GCLM-F: TGTCTTGGAATGCACTGTATCTC
Homo-GCLM-R: CCCAGTAAGGCTGTAAATGCTC
Homo-SLC7A11-F: TCTCCAAAGGAGGTTACCTGC
Homo-SLC7A11-R: AGACTCCCCTCAGTAAAGTGAC
Homo-PTGS2-F: CTGGCGCTCAGCCATACAG
Homo-PTGS2-R: CGCACTTATACTGGTCAAATCCC
Homo-ACSL4-F: CATCCCTGGAGCAGATACTCT
Homo-ACSL4-R: TCACTTAGGATTTCCCTGGTCC
Homo-GPX4-F: GAGGCAAGACCGAAGTAAACTAC
Homo-GPX4-R: CCGAACTGGTTACACGGGAA
Homo-HMOX1-F: AAGACTGCGTTCCTGCTCAAC
Homo-HMOX1-R: AAAGCCCTACAGCAACTGTCG
Homo-ACTB-F: CATGTACGTTGCTATCCAGGC
Homo-ACTB-R: CTCCTTAATGTCACGCACGAT

### Protein extraction and immunoblot analyses

Mouse tissues were homogenized in RIPA lysis buffer (P0013C; Beyotime Biotechnology) in the presence of protease/phosphatase inhibitors (PPC1010-1ML; Sigma-Aldrich) at pH 7.4. The homogenate was centrifuged at 12,000$g$ and 4°C for 15 min. Protein samples were analyzed by Western blot with specific antibodies. Antibodies for actin (66009-1), NQO2 (15767-1-AP), FSP1 (20886-1-AP), and ACSL4 (66617-1-Ig) were from Proteintech. Antibodies for HSP90 (4877S) and SLC7A11 (12691S) were obtained from Cell Signaling Technology. Anti-Flag (HRP) (A8592) was from Sigma-Aldrich. GPX4 (ab125066) and NQO1 (ab80588) were purchased from Abcam.

### Protein purification and activity assays

Expression vectors were transformed into BL-21 competent cells, and LB cultures were inoculated for overnight growth at 37°C while shaking. The next day, the cultures were diluted at 1:100 into 200 ml of LB and incubated at 37°C while shaking until $OD_{600}$ reached 0.6–0.8. Then, the cultures were induced with 0.5 mM isopropyl $\beta$-D-1-thiogalactopyranoside and incubated at 16°C overnight. Bacterial pellets were resuspended in cold lysis buffer containing protease inhibitor cocktail and then applied to ultrasonication. The lysates were centrifuged at 12,000 g and 4°C for 15 min, and the supernatants were applied for His-beads (Beyotime Biotechnology) purification following the manufacturer's instruction. The recombinant proteins were eluted by imidazole and were stored at –80°C until further use.

To measure NADH oxidation kinetics, purified NQO1 was combined with 500 μM NADH and 5–50 μM tanshinone derivatives in a total volume of 100 μl PBS. A reduction in absorbance at 340 nm, corresponding to NADH oxidation, was determined every minute over the course of 1 h. All measurements were taken using BioTek Synergy H1 Hybrid Multi-Mode Reader.

### Thermal stability assays

A total of 2,000,000 HT-1080 cells were seeded into 10-cm dishes. After 1 d, cells were treated with DMSO or 15-047 for 30 min and were harvested by trypsinization. Cells were resuspended in 1 ml of PBS with protease inhibitor cocktail added. Then, cells underwent repeated freezing and thawing in liquid nitrogen and 37°C water bath kettle three times. Cell lysates were centrifuged at 12,000$g$ for 30 min at 4°C. The supernatant was transferred to new PCR tubes at a volume of 100 μl/tube. The PCR tubes were incubated in PCR machines that were set in thermal gradients for 3 min. Then, the lipid was transferred rapidly into new tubes and centrifuged at 21,000$g$ for 60 min at 4°C. Only 40 μl of lipid supernatant was collected into new tubes. The samples were further subjected to label-free proteomic mass spectrometry or SDS–PAGE and Western blot for target proteins.

### Measurement of cellular lipid ROS

A total of 400,000 HT-1080 cells were seeded into six-well plates (Corning). After 1 d, cells were treated with indicated chemicals for the indicated times. After treatment, cells were stained with C11-BODIPY581/591 (D3861; Thermo Fisher Scientific) for 30 min at 37°C and then harvested by trypsinization. Cells were resuspended in PBS and strained through a 40-μm cell strainer (BD Falcon), and then, cells were analyzed using a flow cytometer (Accuri C6; BD Biosciences) equipped with a 488-nm laser for excitation. Data were collected from the FL-1 channel. A minimum of 10,000 cells were analyzed per condition.

### Antioxidant activity measurement

The stable radical 2,2-diphenyl-1-picrylhydrazyl (DPPH) was dissolved in methanol to a final working concentration of 0.05 mM. 1 ml of DPPH solution was added to a small volume (<5 μl) of each test compound dissolved in DMSO, whose final concentration was 0.05 mM. Samples were inverted several times and allowed to incubate at room temperature for 30 min. Samples were then aliquoted to 96-well, solid-bottomed dishes (Corning), and the absorbance at 517 nm was recorded and normalized to background (methanol only).

### Inhibited autoxidation of eggPC liposomes

EggPC liposomes (1 mM) and 581/591 C11-BODIPY (1 μM) were added to a 96-well polypropylene plate (Nunc) in PBS at pH 7.4. This was followed by the addition of either NQO1 (500 nM), NADH (500 μM), tanshinone derivatives (5 μM), or a combination thereof. The plate was incubated for 5 min followed by a vigorous mixing protocol for 5 min at 37°C in the BioTek Synergy H1 plate reader. The plate was

ejected from the plate reader, and the autoxidation was initiated by the addition of 3 μl AAPH (1 mM). Followed by another mixing protocol for 1 min, data were acquired by excitation at 488 nm and emission was measured at 518 nm.

### The docking model of NQO1

The docking model of NQO1 in complex with 15-047 and FAD was generated by Molegro Virtual Docker to illustrate the structure–activity relationship between NQO1 and 15-047. The image was visualized by PyMOL.

### Statistical analysis

The statistical analyses were performed with the GraphPad Prism v.8 software. The statistical significance of data was assessed by a two-tailed $t$ test or two-way ANOVA unless noted otherwise. Data are shown as the mean ± S.D. When the $P$-value is smaller than 0.001, GraphPad Prism v.8 does not provide a precise $P$-value and instead presents these values as $P < 0.001$. $P > 0.05$ was considered not significant. Additional statistical methods are stated in the figure legends.

## Supplementary Information

## Acknowledgements

We Thank Prof. Lu Zhou (Fudan University) for constructive discussion and suggestions and Renke Tan (Fudan University) for the help in compound screening. This work was supported by the NSFC grant (No. 31970684 to H-X Yuan), the National Key Research and Development Project of China (No. 2018YFA0800304 to H-X Yuan), the NSFC grant (No. 21877120 to C Ding), and the Development Fund for Shanghai Talents (No. 2019109 to H-X Yuan). H-X Yuan is also supported by the Innovative Research Team of High-level Local University in Shanghai.

### Author Contributions

T-X Wang: conceptualization, resources, data curation, software, formal analysis, supervision, validation, investigation, visualization, methodology, project administration, and writing—original draft, review, and editing.
K-L Duan: conceptualization, software, formal analysis, supervision, validation, investigation, methodology, project administration, and writing—review and editing.
Z-X Huang: investigation, methodology, and project administration.
Z-A Xue: methodology and project administration.
J-Y Liang: investigation, methodology, and project administration.
Y Dang: resources and methodology.
A Zhang: resources and methodology.
Y Xiong: supervision and writing—review and editing.
C Ding: resources, supervision, and funding acquisition.
K-L Guan: supervision, validation, and writing—review and editing.

H-X Yuan: conceptualization, resources, supervision, funding acquisition, validation, and writing—review and editing.

### Conflict of Interest Statement

K-L Guan is a co-founder of Vivace Therapeutics, and Y Xiong is a co-founder of Cullgen Inc.

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
