## [Reviewer comments · Life Science Alliance]

Life Science Alliance

Tanshinone functions as a coenzyme that confers gain-of-function of NQO1 to suppress ferroptosis

Tian-Xiang Wang, Kun-Long Duan, Zi-Xuan Huang, Zi-An Xue, Jun-Yun Liang, Yongjun Dang, Ao Zhang, Yue Xiong, Chunyong Ding, Kun-Liang Guan, and Hai-Xin Yuan

DOI: <https://doi.org/10.26508/lsa.202201667>

Corresponding author(s): Hai-Xin Yuan, Fudan University and Chunyong Ding, Shanghai Jiao Tong University

Review Timeline:

Submission Date:	2022-08-12
Editorial Decision:	2022-10-10
Revision Received:	2022-10-19
Editorial Decision:	2022-10-19
Revision Received:	2022-10-20
Accepted:	2022-10-20

Transaction Report:

October 10, 2022

Re: Life Science Alliance manuscript #LSA-2022-01667-T

Dr. Hai-Xin Yuan
Fudan University
The Molecular and Cell Biology Laboratory
Shanghai 20032
China

Dear Dr. Yuan,

Thank you for submitting your manuscript entitled "Tanshinone functions as a coenzyme that confers gain-of-function of NQO1 to suppress ferroptosis" to Life Science Alliance. The manuscript was assessed by expert reviewers, whose comments are appended to this letter. We invite you to submit a revised manuscript addressing the Reviewer comments.

Thank you for this interesting contribution to Life Science Alliance. We are looking forward to receiving your revised manuscript.

Sincerely,

B. MANUSCRIPT ORGANIZATION AND FORMATTING:

Reviewer #1 (Comments to the Authors (Required)):

This manuscript by Wang et al describes the isolation of tanshinones as natural products that can inhibit ferroptosis, a form of non-apoptotic cell death, in combination with the enzyme NQO1. Results from both cell-based and animal models are presented and together make a compelling argument in favor the molecular model, which in turn may explain the purported therapeutic activity of tanshinones. This is a straightforward paper with interesting results and conclusions. My comments are mostly minor:

-In the Discussion the authors claim that the tanishones are the most potent ferroptosis inhibitors known to date. This seems like a bold assertion in the absence of direct comparative studies, with many other molecules being reported to have low nanomolar potency against ferroptosis (e.g., see the work of Derek Pratt, e.g., PMID: 35921655). It may be safer to simply note that these are among the most potent.

-Figure legends could use clearer details about the concentration of tanshinones used in specific experiments, like Figure 1B and 1D, where no concentration information for the inhibitor is given in the Figure panel or legend.

-Some experiments (e.g., Figure 3C) appear to have been performed only a single time, and in other cases it is not clear whether the data are derived from experiments performed on different days or merely technical replicate wells from one experiment.

-Molecular weight markers should be added to all western blots in Figure 5.

-Line 46 and elsewhere. No need for lipid ROS to be hyphenated.

-Line 83. RTA is radical trapping antioxidant, not radical trapping agent.

-Line 199. Not clear why the Yang et al., 2014 paper is cited in connection with I/R injury. That paper appears to deal exclusively with cancer.

-The text needs some editing for language throughout. Line 64, for example, should probably read "Taking advantage..." not "Take advantage...". This is just one of many examples.

Reviewer #2 (Comments to the Authors (Required)):

In the present manuscript, Wang et al provide a very careful characterization of the mode of action of Tanshinone and its derivatives in preventing ferroptosis. The authors demonstrate that Tanshinone are extremely potent ferroptosis inhibitors both in vivo and in vitro. They proceed to show using a series of assays that the quinone oxidoreductase (NQO1) is required to reduce Tanshinone to its radical trapping-active metabolite. The model proposed are validated using a combination of biochemical assays and validated using NQO1 knockout cell lines and mice.

The conclusion are sounds and of general interest. The work is of high quality and the data well presented.

I only apologize the authors and the editors for the delay in reviewing this piece mostly because I have nothing to add to this very nice study.

José Pedro Friedmann Angeli

Reviewer Comments:

Reviewer #1 (Comments to the Authors (Required)):

This manuscript by Wang et al describes the isolation of tanshinones as natural products that can inhibit ferroptosis, a form of non-apoptotic cell death, in combination with the enzyme NQO1. Results from both cell-based and animal models are presented and together make a compelling argument in favor the molecular model, which in turn may explain the purported therapeutic activity of tanshinones. This is a straightforward paper with interesting results and conclusions. My comments are mostly minor:

Response: We appreciate the reviewer's positive view and constructive comments to improve the quality of this manuscript.

Major comments:

(1) -In the Discussion the authors claim that the tanshinones are the most potent ferroptosis inhibitors known to date. This seems like a bold assertion in the absence of direct comparative studies, with many other molecules being reported to have low nanomolar potency against ferroptosis (e.g., see the work of Derek Pratt, e.g., PMID: 35921655). It may be safer to simply note that these are among the most potent.

Response: We have revised related statement to "The low nanomolar EC50 of tanshinone in suppressing ferroptosis makes it among the most potent ferroptosis inhibitors known to date".

(2) -Figure legends could use clearer details about the concentration of tanshinones used in specific experiments, like Figure 1B and 1D, where no concentration information for the inhibitor is given in the Figure panel or legend.

Response: Thanks for the comments. In the Figure legends for Figure 1B, we have added the concentration of tanshinone compounds used in taking representative cell morphology (50 nM). In figure 1D, the different concentrations of tanshinone compounds have already been indicated in the figure panel.

(3) -Some experiments (e.g., Figure 3C) appear to have been performed only a single time, and in other cases it is not clear whether the data are derived from experiments performed on different days or merely technical replicate wells from one experiment.

Response: As for the concentration-dependent compounds treatment experiments like Figure 3C and other EC50 values measurement experiments, data are presented as mean \pm S.D., n = 3 independent repeats. EC50 values were calculated by nonlinear

regression [Log(agonist) vs. response-Variable slope (four parameters)]. We have stated this clearly in the figure legends.

(4) -Molecular weight markers should be added to all western blots in Figure 5.

Response: The molecular weight markers for all western blots have been added in Figure 5.

(5) -Line 46 and elsewhere. No need for lipid ROS to be hyphenated.

Response: The hyphen between lipid and ROS are deleted.

(6) -Line 83. RTA is radical trapping antioxidant, not radical trapping agent.

Response: We have revised this inaccurate description in the manuscript. We thank the Reviewer for pointing out our mistake and corrected this in our revised manuscript.

(7) -Line 199. Not clear why the Yang et al., 2014 paper is cited in connection with I/R injury. That paper appears to deal exclusively with cancer.

Response: This was a mistake. We have replaced the citation of Yang et al., 2014 with Friedmann Angeli *et al.*, 2014 and Gao *et al.*, 2015.

(8) -The text needs some editing for language throughout. Line 64, for example, should probably read "Taking advantage..." not "Take advantage...". This is just one of many examples.

Response: We apologize for the language problem. We have carefully proofread the manuscript to minimize typographical, grammatical, and bibliographical errors and have sent the manuscript to AJE company to polish the language.

Response:

Reviewer #2 (Comments to the Authors (Required)):

In the present manuscript, Wang et al provide a very careful characterization of the mode of action of Tanshinone and its derivatives in preventing ferroptosis. The authors demonstrate that Tanshinone are extremely potent ferroptosis inhibitors both in vivo and in vitro. They proceed to show using a series of assays that the quinone oxidoreductase (NQO1) is required to reduce Tanshinone to its radical trapping-active metabolite. The model proposed are validated using a combination of biochemical assays and validated using NQO1 knockout cell lines and mice. The conclusion are sounds and of general interest. The work is of high quality and the data well presented.

I only apologize the authors and the editors for the delay in reviewing this piece mostly because I have nothing to add to this very nice study.

Response: We appreciate the Reviewer for the general enthusiasm and positive evaluation of our manuscript.

October 19, 2022

RE: Life Science Alliance Manuscript #LSA-2022-01667-TR

Dr. Hai-Xin Yuan
Fudan University
131 Dong'an Road
Shanghai 20032
China

Dear Dr. Yuan,

Thank you for submitting your revised manuscript entitled "Tanshinone functions as a coenzyme that confers gain-of-function of NQO1 to suppress ferroptosis". We would be happy to publish your paper in Life Science Alliance pending final revisions necessary to meet our formatting guidelines.

- please add ORCID ID for both corresponding authors-you should have received instructions on how to do so
- please add the Twitter handle of your host institute/organization as well as your own or/and one of the authors in our system
- please add a callout for Figure S5C to your main manuscript text

Figure Check:

- the scale bars in Figure 1B are hard to see

A. FINAL FILES:

B. MANUSCRIPT ORGANIZATION AND FORMATTING:

**Submission of a paper that does not conform to Life Science Alliance guidelines will delay the acceptance of your

manuscript.**

The license to publish form must be signed before your manuscript can be sent to production. A link to the electronic license to publish form will be sent to the corresponding author only. Please take a moment to check your funder requirements.

Sincerely,

October 20, 2022

RE: Life Science Alliance Manuscript #LSA-2022-01667-TRR

Dr. Hai-Xin Yuan
Fudan University
131 Dong'an Road
Shanghai 20032
China

Dear Dr. Yuan,

Thank you for submitting your Research Article entitled "Tanshinone functions as a coenzyme that confers gain-of-function of NQO1 to suppress ferroptosis". It is a pleasure to let you know that your manuscript is now accepted for publication in Life Science Alliance. Congratulations on this interesting work.

DISTRIBUTION OF MATERIALS:

Again, congratulations on a very nice paper. I hope you found the review process to be constructive and are pleased with how the manuscript was handled editorially. We look forward to future exciting submissions from your lab.

Sincerely,
